# Not All Samples Are Equal: Quantifying Instance-level Difficulty in Targeted Data Poisoning

## Abstract

Targeted data poisoning attacks pose an increasingly serious threat due to their ease of deployment and high success rates. These attacks aim to manipulate the prediction for a single test sample in classification models. Unlike indiscriminate attacks that aim to decrease overall test performance, targeted attacks present a unique threat to individual test instances. This threat model raises a fundamental question: *what factors make certain test samples more susceptible to successful poisoning than others?* We investigate how attack difficulty varies across different test instances and identify key characteristics that influence vulnerability. This paper introduces three predictive criteria for targeted data poisoning difficulty: ergodic prediction accuracy (analyzed through clean training dynamics), poison distance, and poison budget. Our experimental results demonstrate that these metrics effectively predict the varying difficulty of real-world targeted poisoning attacks across diverse scenarios, offering practitioners valuable insights for vulnerability assessment and understanding data poisoning attacks.

## 1 Introduction

In the past decade, machine learning (ML) models have achieved great success in various domains, largely due to the vast amount of training data available on the internet. However, this reliance on massive training datasets not only increases computational costs but also introduces significant security vulnerabilities during the data collection process (Szegedy et al., 2014; Kumar et al., 2020). Adversaries can exploit these vulnerabilities through data poisoning attacks which deliberately inject malicious samples into training data either actively or passively (Gao et al., 2020; Wakefield, 2016; Shejwalkar et al., 2022; Lyu et al., 2020; Carlini et al., 2024). These attacks are particularly concerning because they can compromise model integrity at its foundation, affecting all downstream applications and users of the poisoned model (Goldblum et al., 2023).

Targeted data poisoning attacks represent a specialized form of this threat, where attackers aim to manipulate model behavior for specific test instances while maintaining normal performance on all other inputs (e.g., Shafahi et al., 2018; Aghakhani et al., 2021; Guo & Liu, 2020; Zhu et al., 2019; Geiping et al., 2021). We primarily focus on classification models (and briefly discuss generative models in Appendix E), where the objective is to misclassify a particular sample to a predetermined class while maintaining correct predictions for all other inputs. Such attacks are difficult to detect as they leave little evidence in overall model performance metrics.

Current evaluations of targeted attack threats typically rely on randomly selected test samples to report overall attack success rates (e.g., Aghakhani et al., 2021; Geiping et al., 2021) as an average assessment. However, our observations reveal substantial variation in attack effectiveness across different test instances, with no clear understanding of what characteristics drive these disparities. This paper addresses this critical gap by investigating two key research questions: *(1) what factors determine why a certain test sample is more vulnerable to targeted poisoning attacks than others? and (2) can we develop reliable metrics to predict the difficulty of poisoning a specific instance?*

We address the first question by identifying three critical factors that influence poisoning difficulty: the inherent classification difficulty during clean training, the distance in model parameter space required to achieve poisoning, and the attacker's resource constraints measured by poison budget.

To quantify these factors, we introduce three corresponding metrics that naturally predict poisoning difficulty: (1) ergodic prediction accuracy (EPA) derived from clean training dynamics, (2) poisoning distance $\delta$, and (3) poisoning budget lower bound $\tau$. Importantly, all our proposed metrics can be calculated *using only clean training data and processes*, without requiring the simulation of actual poisoning attacks—making them practical and accessible tools for defenders to assess vulnerability and prioritize protection efforts.

Our experimental results confirm that all three proposed factors strongly correlate with real-world attack performance. The metrics we developed effectively predict poisoning difficulty across various test instances, capturing different dimensions and levels of vulnerability. Specifically, we find that ergodic prediction accuracy (EPA) serves as a powerful indicator for distinguishing between easy-to-poison and hard-to-poison test samples. Meanwhile, poisoning distance $\delta$ and budget lower bound $\tau$ provide more fine-grained predictions for specific poison classes from complementary perspectives. Together, these three metrics form a comprehensive framework that enables defenders to assess poisoning vulnerability for any given test sample.

In summary, our work makes three distinct contributions: (1) We identify classification difficulty during clean training, parameter-space poisoning distance, and poison budget as the key factors determining targeted poisoning vulnerability; (2) We introduce three corresponding metrics: ergodic prediction accuracy (EPA), poisoning distance, and budget lower bound, all calculable using only clean training processes without any expensive attack and retraining; (3) Our experiments demonstrate the effectiveness of these metrics and introduce a framework that enables defenders to clearly identify and understand instance-level difficulty in targeted data poisoning.

## 2 BACKGROUND

**Threat model and notations:** We first specify our threat model and list our notations below.

- **Objective**: We consider an adversary who tries to *alter the prediction* of a specific test sample $\mathbf{x}_t$[1] from the correct class $y_t$ (or target class) to a specific poison class $y_p$.

- **Attack deployment**: The attacker reaches the objective by injecting a poisoned set $\mathcal{D}_p$ into the clean training set $\mathcal{D}_c$. We assume the defender is training a machine learning model on the merged dataset, i.e., $\mathcal{D}_{tr} = \mathcal{D}_c \cup \mathcal{D}_p$ and deploy the model on a test set $\mathcal{D}_{test}$ that contains $\mathbf{x}_t$.

- **Attacker's knowledge**: We consider a *white-box attack*,[2] where the attacker is aware of the clean training set $\mathcal{D}_c$, the machine learning model architecture and the training scheme utilized by the defender, and the inclusion of the test sample $\mathbf{x}_t$ in the test set.

- **Attacker's budget & Constraint**: We define the attacker's budget as $\varepsilon = \frac{|\mathcal{D}_p|}{|\mathcal{D}_{tr}|}$, i.e., the (relative) percentage of poisoning data. The budget for targeted attacks is usually low, e.g., $\varepsilon = 1\%$. We also set constraints on $\mathcal{D}_p$, where it only contains "clean-labeled" poison data, namely that elements in $\mathcal{D}_p$ are generated by adding human imperceptible noises (e.g., with $\ell_\infty$ constraints) to clean training images without changing their original labels.

- **Attack evaluation**: We define an attack to be successful when the prediction of the target sample using the poisoned model $f(\mathbf{x}_t; \mathbf{w}_p)$ (see notation below) is equal to the poisoned label $y_p$. Note that attack success is strictly stronger than misclassification.

**Other notation:** We denote the clean model parameters (a model $f$ trained only on $\mathcal{D}_c$) to be $\mathbf{w}_c$ and poisoned model parameters to be $\mathbf{w}_p$. Let $\ell(\mathbf{w}, \mathbf{z})$ be our loss that measures the fitness of our model $\mathbf{w}$ on data $\mathbf{z} \in \mathbb{Z}$, e.g., $\mathbf{z} = (\mathbf{x}, y)$. Let $\mathbf{g}(\mathbf{z}) = \mathbf{g}(\mathbf{z}; \mathbf{w}) = \nabla_{\mathbf{w}} \ell(\mathbf{z}; \mathbf{w})$ be the gradient vector with respect to a fixed model $\mathbf{w}$ evaluated at the data $\mathbf{z}$.

---

[1]We note that the attacker cannot change $\mathbf{x}_t$, but indirectly change the model's behavior by deploying $\mathcal{D}_p$ through retraining. We highlight this is a key difference from adversarial examples, which directly modifies $x_t$ without any poisoned set or retraining.

[2]We note that the attacks could possibly be performed in a partially *black-box* fashion, where the attackers need to apply surrogate models and training procedure. However, such attacks suffer a severe performance drop (Schwarzschild et al., 2021). To ensure the strongest threat is measured from the defender's perspective to ensure maximum security, we consider the *white-box* setting instead.

**Targeted data poisoning:**    In this paper, we focus on targeted data poisoning attacks (e.g., Shafahi et al., 2018; Aghakhani et al., 2021; Guo & Liu, 2020; Zhu et al., 2019) that affect only specific test samples and discuss other types of poisoning attacks in Appendix B.1. Given a test sample $(\mathbf{x}_t, y_t)$, the problem can be formulated into the bi-level optimization problem below:

$$\min_{\mathcal{D}_p} \ell((\mathbf{x}_t, y_p), \mathbf{w}_*), \text{ s.t. } \mathbf{w}_* \in \operatorname*{argmin}_{\mathbf{w}} \ell(\mathcal{D}_c \cup \mathcal{D}_p, \mathbf{w}),$$

where the attacker aims to enforce the prediction of $x_t$ to be $y_p$ through crafting and injecting $\mathcal{D}_p$ into the training set $\mathcal{D}_{tr}$. This problem is hard to solve directly as the outer maximization problem depends on $\mathcal{D}_p$ only implicitly through the solution $\mathbf{w}_*$ of the inner problem. Existing attacks consider relaxations of this primal problem, for example, a fixed feature extractor (Aghakhani et al., 2021; Zhu et al., 2019) or approximating the gradient of target parameters (Shafahi et al., 2018). As our poisoning difficulty metrics do not depend on the specific design of attack algorithms, we omit the attack details and refer readers to the above references. We note that data poisoning attacks can be further classified according to their attack scheme, specifically, the construction of $\mathcal{D}_{tr}$, and we extend more discussion in Appendix B.2. Targeted attacks are insidious as they do not cause significant performance degradation (hence harder to detect) while still capable of causing system failure on targeted test instances. Previous works (Shafahi et al., 2018; Geiping et al., 2021; Zhu et al., 2019) have demonstrated the efficacy of such attacks against deep neural networks, reporting high attack success rates. However, these reported success rates are typically calculated by averaging over randomly selected test samples (Schwarzschild et al., 2021; Geiping et al., 2021). This aggregated metric fails to capture the instance-level difficulty of targeted data poisoning attacks, a critical gap we aim to address in this work.

## 3    DIFFICULTY OF TARGETED DATA POISONING

To quantify the variance of attack performance on different test instances, we perform an experiment by applying a state-of-the-art attack called gradient matching (Geiping et al., 2021). We choose the first 100 test samples (based on target id) in the class "plane" of the CIFAR-10 dataset (Krizhevsky, 2009) and perform gradient matching to classify them into the nine other (poison) classes (900 attacks in total). For each attack, we perform 8 independent trials with different model initialization and report the attack success rate, calculated over the 8 independent runs, as a histogram for all 900 attacks in Figure 1. Our results show a high variance in the distribution of attack success rate, which highlights the necessity of understanding instance-level poisoning difficulty. We will show in Section 4 that the choice of the target class also introduces high variance.

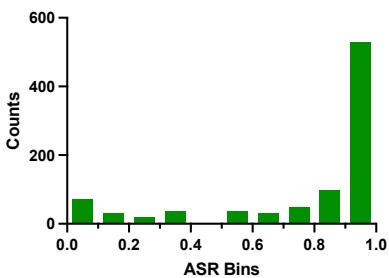

Figure 1: Histogram of the attack success rate of gradient matching over 8 runs on attacking 100 test samples in the class "plane" in CIFAR-10.

In this section, we introduce novel tools and metrics for understanding and quantifying the difficulty of targeted data poisoning. A key strength of our approach is its generality: our tools operate *independently* of specific data poisoning attack implementations and rely solely on *clean-training* dynamics (and the test set we examine). This attack-agnostic framework enables defenders to assess vulnerability without requiring knowledge of or access to particular attack methodologies. Recall that we denote a clean model as $\mathbf{w}_c$, which we will frequently use in this section.

### 3.1    POISONING DIFFICULTY PREDICTION WITH CLEAN TRAINING DYNAMICS

Naively, for a linear model such as a support vector machine (SVM), the prediction of a test sample positioned near the decision boundary is likely to be altered by small changes to the model. One might therefore consider using the distance between a test sample and the decision boundary to determine its robustness, or equivalently, its poisoning difficulty. However, this approach falls short for neural networks and fails to account for the complex training dynamics of non-linear models.

To study the poisoning difficulty of neural networks, we propose an intuitive hypothesis, motivated by the linear example above:

**Hypothesis A.** *The classification difficulty of a test sample $\mathbf{x}_t$ is negatively correlated with its poisoning difficulty, i.e., a sample $\mathbf{x}_t$ that is easy to classify is correspondingly difficult to poison.*

To verify the above hypothesis, it is necessary to establish a robust measure of the difficulty of classification, which we approach by examining the prediction correctness throughout training:

**Definition 1** (Ergodic Prediction Accuracy, EPA). *We say the classification difficulty for a target sample $\mathbf{x}_t$ can be measured by the ergodic average correctness (denoted by the indicator function) for $N$ training epochs with $M$ different initializations:*

$$\text{EPA} = \frac{1}{MN}\sum_{m=1}^{M}\sum_{n=1}^{N}\mathbb{1}\{f_{m,n}(\mathbf{x}_t) = y_t\},$$

*i.e., a model with higher EPA is easier to classify, thus harder to attack, and vice versa.*

When the model update is ergodic (Royden & Fitzpatrick, 2010) and with large $M$ and $N$, EPA converges to $\Pr[f(\mathbf{x}_t; \mathbf{w}^*) = y_t]$ where $\mathbf{w}^*$ follows the invariant distribution of the update process.

Although we use the prediction of the model $f_{m,n}(\mathbf{x}_t)$ above, EPA can also be calculated using the model's confidence (i.e., logits). We demonstrate the performance of both approaches in Section 4. We acknowledge that EPA represents just one method for measuring classification difficulty, and we discuss alternative approaches in Appendix B.3.

While EPA provides a convenient measure, it does not fully capture a test sample's robustness against attacks targeting a specific poisoned label $y_p$. We observe significant variations across different choices of $y_p$ as demonstrated in Figure 2. Furthermore, EPA is calculated solely based on clean training and does not account for realistic attack scenarios.

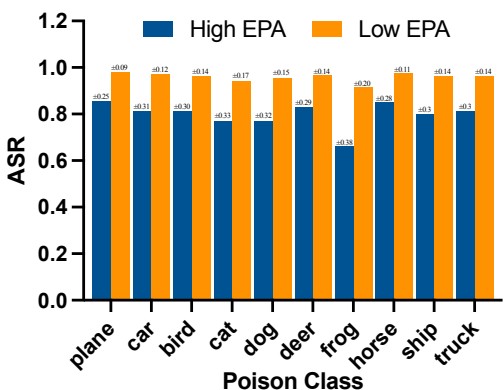

Figure 2: The attack success rates of gradient matching on CIFAR-10 on different poison classes $y_p$ for 100 test samples $\mathbf{x}_t$ with the highest/lowest EPA (50 for each group). We show the average attack success rate for both groups across all 10 possible poison classes.

### 3.2 POISONING DIFFICULTY PREDICTION WITH POISONING DISTANCE

To address the above limitations of EPA, we provide an alternative tool called *poisoning distance* to measure poisoning difficulty, which takes the choice of $y_p$ into account. Specifically, we consider the ultimate goal of targeted poisoning:

***Goal 1****: An adversary aims at modifying model parameters from clean parameters $\mathbf{w}_c$ to poisoned ones $\mathbf{w}_p$ such that $f(\mathbf{x}_t; \mathbf{w}_p) = y_p$.*

Data poisoning implements an indirect way to achieve this goal through crafting $\mathcal{D}_p$ and training on $\mathcal{D}_c \cup \mathcal{D}_p$. Goal 1 enables us to measure poisoning difficulty by comparing $\mathbf{w}_p$ and $\mathbf{w}_c$ directly. Specifically, we propose a hypothesis on poisoning distance:

**Hypothesis B.** *The distance $\delta = d(\mathbf{w}_c, \mathbf{w}_p)$ between a clean model $\mathbf{w}_c$ and a (targeted) poisoned model $\mathbf{w}_p$ is positively correlated with poisoning difficulty, i.e., a test sample $\mathbf{x}_t$ with higher $\delta$ is correspondingly more difficult to poison.*

Note that $d(\cdot)$ is a distance function that we will specify soon. Here $\delta$ is a sample-wise metric as its calculation depends on the sample-specific poisoned parameter $\mathbf{w}_p$. Moreover, Hypothesis B naturally considers the choice of $y_p$, which is embedded in the definition of $\mathbf{w}_p$.

However, from a defender's perspective, validating Hypothesis B directly is non-trivial as the calculation of $\delta$ depends on the poisoned parameters $\mathbf{w}_p$, which are unknown without performing an actual attack. Luckily, data poisoning is not the only viable way to achieve Goal 1, and we propose a proxy to generate $\mathbf{w}_p$ and measure $\delta$ without performing any data poisoning attacks:

**Definition 2** (Poisoning Distance $\delta$)**.** *Starting from a clean model* $\mathbf{w}_c$, *we say the poisoning distance is the smallest step size required to modify* $\mathbf{w}_c$ *in one step such that the model classifies* $\mathbf{x}_t$ *as* $y_p$:

$$\delta = \min\{\eta > 0 : \tilde{f}(\mathbf{x}_t; \mathbf{w}_c - \eta \cdot \mathbf{g}) = y_p\},$$

*where* $\mathbf{g} = \nabla_{\mathbf{w}} \ell(\tilde{f}(\mathbf{x}_t; \mathbf{w}_c), y_p)$. *Naturally, we also obtain our proxy of* $\mathbf{w}_p = \mathbf{w}_c - \eta \cdot \mathbf{g}$.

While such a proxy may be different from a real data poisoning attack, $\delta$ intuitively measures the efforts needed to achieve the attack goal from a gradient perspective[3].

Of course, there exist various algorithms for finding $\delta$. In this paper, we provide a simple $\delta$ estimator in Algorithm 1 in Appendix C using binary search. Here we highlight that one advantage of the estimation of $\delta$ over EPA is that it does not depend on the training process or the clean data $\mathcal{D}_c$, which is extremely handy for users that has access only to the model weights (which is very common for foundation models) to quantify the poisoning difficulty of their own personal data.

### 3.3 POISONING DIFFICULTY PREDICTION WITH POISON BUDGET

Aside from classification difficulty and poisoning distance, an alternative way to measure poisoning difficulty is through the poison budget $\varepsilon$. It is clear that an attack is easier if less poisoned data is required, i.e., a lower $\varepsilon$ suggests an easier attack. Here we aim to answer an intriguing question:

*Is it possible to measure the lowest* $\varepsilon$ *needed to poison a model such that a given test sample* $\mathbf{x}_t$ *is misclassified as* $y_p$, *without performing any attacks?*

Conveniently, Lu et al. (2023) provides valuable theoretical tools for measuring the (relative) number of poisoned samples $|\mathcal{D}_p|$ needed to reach some target parameters $\mathbf{w}_p$ (e.g., the proxy we generated in Definition 2), i.e., the role of poison budget $\varepsilon$. Specifically, Lu et al. (2023) provides a lower bound (or necessary condition) with respect to $\varepsilon$ on poisoning reachability. Without diving into all technical details and derivations, we directly present a simplified version of their results:

**Theorem 1** (Poisoning reachability, Theorem 2 of Lu et al. (2023))**.** *Given a classification task with* $c$ *classes and a set of target parameters* $\mathbf{w}_p$, $\mathbf{w}_p$ *is poisoning reachable (defined by vanishing gradient over training on* $\mathcal{D}_c \cup \mathcal{D}_p$) *only if the condition below holds (necessary condition):*[4]

$$\varepsilon \geq \tau := \max\left\{ \frac{\langle \mathbf{w}_p, \mathbf{g}(\mathcal{D}_c) \rangle}{\mathscr{W}(c - 1/e)}, 0 \right\},$$

*where* $\mathscr{W}(\cdot)$ *is Lambert's W function,* $\mathbf{g}(\mathcal{D}_c) = \mathbf{g}(\mathcal{D}_c; \mathbf{w}_p) = \frac{1}{|\mathcal{D}_c|} \sum_{\mathbf{z} \in \mathcal{D}_c} \nabla_{\mathbf{w}_p} \ell(\mathbf{z}; \mathbf{w}_p)$.

Theorem 1 enables us to calculate $\tau$, the lower bound of poisoning budget $\varepsilon$ for a given target test sample $\mathbf{x}_t$, the corresponding poison class $y_p$, the target parameter $\mathbf{w}_p$, and the clean training set $\mathcal{D}_c$, In Section 4, we will show that $\tau$ is a direct indicator of poisoning difficulty.

## 4 EXPERIMENTS

In this section, we (1) introduce our experimental settings; (2) present our results on poisoning difficulty prediction using EPA; (3) demonstrate the effectiveness of the poisoning distance $\delta$ and poisoning reachability $\tau$ on measuring poisoning difficulty; (4) show our ablation studies on datasets, model architectures, and poisoning budget.

### 4.1 EXPERIMENTAL SETTINGS

We mainly examine the targeted attacks in the unified benchmark provided in Schwarzschild et al. (2021)[5]. We do not consider backdoor attacks as they require trigger injection during inference and the attack difficulty highly depends on the trigger, And thus are beyond the scope of this paper.

---

[3]We note that our core idea of poisoning distance is closely related to model-targeted indiscriminate attacks and we extend our discussion in Appendix B.4.

[4]Note that Theorem 2 in Lu et al. (2023) presents poisoning reachability for binary linear models, and we consider the general form in Equation (10) on multiclass neural networks.

[5]https://github.com/aks2203/poisoning-benchmark

**Datasets & models:** We consider classification tasks on CIFAR-10 (Krizhevsky, 2009) with 10 classes, 50000 clean training samples and 10000 test samples in our main experiments, and Tiny-ImageNet (Le & Yang, 2015) with 200 classes, 100000 clean training samples, 10000 validation samples, and 10000 test samples in our ablation study. We apply ResNet-18 (He et al., 2016) for CIFAR-10 and VGG-16 (Simonyan & Zisserman, 2014) for TinyImageNet. **Training schemes:** We consider two training schemes: (1) Training from scratch, where we initialize the model with random weights. For clean training, we use the clean training set $\mathcal{D}_c$, for data poisoning we use $\mathcal{D}_c \cup \mathcal{D}_p$[6]; (2) Transfer learning for CIFAR-10, where we utilize a frozen model pretrained on CIFAR-100, and fine-tune an additional linear head on a subset of CIFAR-10 that contains the first 250 images per class. For both scenarios, we train the model for 40 epochs.

**Targeted attacks:** We examine three attack methods listed in the unified benchmark: (1) Gradient matching (GM) (Geiping et al., 2021)[7] for training from scratch; (2) Feature collision (FC) (Shafahi et al., 2018) for transfer learning; and (3) Bullseye polytope (BP) (Aghakhani et al., 2021) for both.[8] For training from scratch, we perform 8 random model initializations and calculate the attack success rate (ASR) by dividing the number of successful attacks by 8. For transfer learning, as the model initialization is mostly fixed, we only consider one attack trial each. For all attacks, unless specified otherwise, we use a poisoning budget $\varepsilon = 1\%$.

**Measuring poisoning difficulty:** (1) To calculate EPA for each test sample, we train the model with $M = 100$ (CIFAR-10), and $M = 8$ (TinyImageNet) random initializations for $N = 40$ epochs. We consider the model prediction for training from scratch and model confidence for transfer learning; (2) To obtain $\delta$, for each choice of $(\mathbf{x}_t, y_p)$, we consider 8 model initializations to generate 8 $\mathbf{w}_c$, apply Algorithm 1 on each $\mathbf{w}_c$ and obtain the average; (3) For the calculation of $\tau$, we only apply one set of $\mathbf{w}_c$ and consider the number of classes $c = 10$ for CIFAR-10 and $c = 200$ for TinyImageNet and apply Theorem 1. We further report our resource and computational time in Appendix D.1.

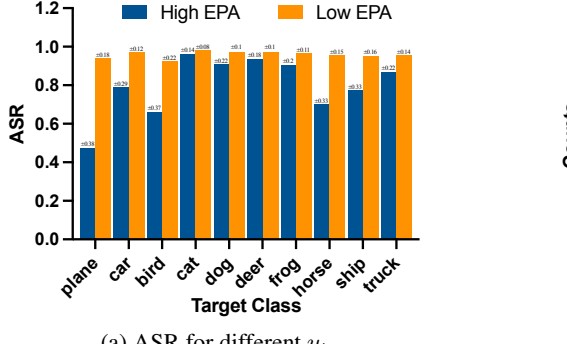

(a) ASR for different $y_t$

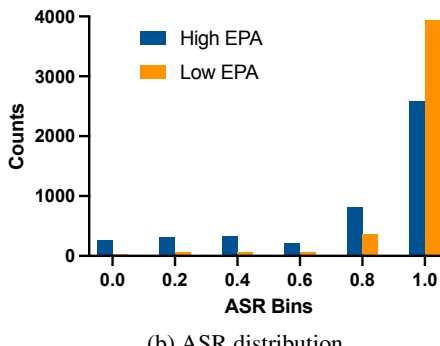

(b) ASR distribution

Figure 3: Measuring the poisoning difficulty of GM on CIFAR-10 (training from scratch) using EPA. We plot the ASR for low/high EPA test samples in each target class $y_t$ in (a), and the overall ASR distribution as a histogram in (b).

### 4.2 POISONING DIFFICULTY PREDICTION WITH EPA

Recall in Section 3.1 Hypothesis A we hypothesize a negative correlation between classification difficulty and poisoning difficulty. In this section, we aim to show that (1) EPA, defined in Definition 1, is a good indicator of poisoning difficulty; (2) EPA is not capable of differentiating between different poison classes $y_p$ for a test sample $x_t$, and is ineffective on further ranking the poisoning difficulty within groups of targets with similar EPA.

**Training from scratch:** For the from-scratch setting on ResNet-18/CIFAR-10, we perform clean training on $\mathcal{D}_c$ with the prespecified $M$ and $N$ to identify the 50 target samples with the highest and lowest EPA in each target class $y_t$. For each target sample, we perform the GM attack on all possible

---

[6]Note that for replacing attacks, $\mathcal{D}_c$ can be changed after poisoning, see Section B.2 for discussion.

[7]https://github.com/JonasGeiping/poisoning-gradient-matching

[8]We neglect Convex Polytope (CP) (Zhu et al., 2019) as it is extremely expensive. It takes 100 seconds and 40 seconds to run one attack instance for BP and FC, respectively, while it takes more than 1 hour to run CP on a NVIDIA 4090 GPU. As our experiments require thousands of attack instances, it is infeasible to run CP.

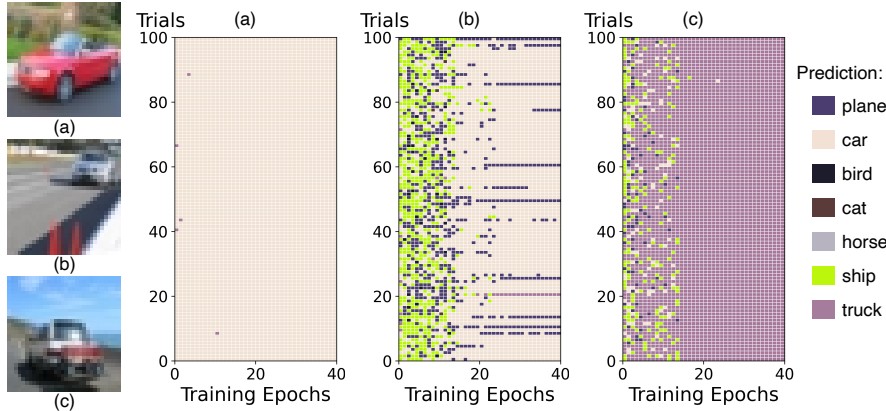

Figure 4: `EPA` for three test instances in the class "car". Image (a): high `EPA`: 0.9988; ASR: 22.22%. Image (b): medium `EPA`: 0.6775; ASR: 90.28%. Image (c): low `EPA`: 0.0275; ASR: 98.61%.

Table 1: The ASR, change of confidence for $y_p$ and $y_t$ before/after attack for high/low `EPA` test samples with FC and BP attack on CIFAR-10 with transfer learning.

| $\mathbf{x}_t$ | ASR | | change of confidence ($y_p$) | | change of confidence ($y_t$) | |
|---|---|---|---|---|---|---|
| | FC | BP | FC | BP | FC | BP |
| high `EPA` | 0.012 | 0.498 | $0.040 \pm 0.031$ | $0.479 \pm 0.133$ | $-0.048 \pm 0.036$ | $-0.503 \pm 0.137$ |
| low `EPA` | 0.284 | 0.947 | $0.156 \pm 0.041$ | $0.661 \pm 0.063$ | $-0.198 \pm 0.053$ | $-0.750 \pm 0.058$ |

(9) poison classes $y_p$ for 8 randomly initialized model weights. We thus run $(50 + 50) \times 10 \times 9 \times 8 = 72000$ attack instances in total. We report the computation time for each attack instance in Section D.1. We present our main results in Figure 3 and observe that `EPA` is a reliable indicator of poisoning difficulty, where a higher `EPA` suggests much lower ASR on average. We note that Figure 3(a) demonstrates a discrepancy between different target classes, specifically, for classes cat and dog, the ASR differences are relatively small between high `EPA` test samples and low `EPA` ones. We argue that these classes are generally easier to poison as they are more difficult to classify, with 82.66% (cat) and 85.73% (dog) clean test accuracy, while that of all other classes is greater than 90%. We present a visualization of `EPA` for instances with high, medium, and low `EPA` values in Figure 4, demonstrating its effectiveness as an indicator of instance-level poisoning difficulty.

**Transfer learning:** For the transfer learning setting on ResNet-18 and CIFAR-10, we again identify the 50 target samples with the highest and lowest `EPA` and apply an additional restriction that all identified target images are classified correctly at the final epoch in all $M$ clean training runs. Table 1 shows our main result on CIFAR-10 for two attacks FC and BP. We observe that the average ASR for test samples with high `EPA` is much lower than the ones with low `EPA` for both attacks. Additionally, as we start from a pre-trained model with reasonable performance, it is interesting to visualize the model change after an attack. Thus, we report the average change of confidence for $y_p$ and $y_t$ for each test sample and confirm that `EPA` is a reliable metric to measure poisoning difficulty. Moreover, to check whether `EPA` is a reliable tool for predicting attack success, we report the average `EPA` of test targets that are successfully and unsuccessfully poisoned in Table 2. We observe that `EPA` is capable of clearly differentiating between successful attacks and failed attacks in most cases, while the prediction region may occasionally overlap.

Table 2: The average `EPA` and confidence of $y_p$ after clean training for successful/failed attacks using the FC and BP attack on CIFAR-10 with transfer learning.

| Attack Success | Average `EPA` | | Confidence of $y_p$ | |
|---|---|---|---|---|
| | FC | BP | FC | BP |
| ✓ | $0.411 \pm 0.116$ | $0.589 \pm 0.255$ | $0.142 \pm 0.092$ | $0.043 \pm 0.074$ |
| ✗ | $0.725 \pm 0.264$ | $0.912 \pm 0.148$ | $0.012 \pm 0.033$ | $0.001 \pm 0.002$ |

Table 3: Measuring the poisoning difficulty of GM on CIFAR-10 using $\delta$ and $\tau$ for the poison classes with the highest and lowest ASR over all target classes $y_t$. Anomaly cases where the prediction does not conform to ASR are marked with underline.

| $y_t$ | Lowest ASR $y_p$ | | | Highest ASR $y_p$ | | |
|---|---|---|---|---|---|---|
| | distance $\delta$ | budget $\tau$ | ASR | distance $\delta$ | budget $\tau$ | ASR |
| plane | $0.119 \pm 0.042$ | $0.00237 \pm 0.00336$ | $0.57 \pm 0.42$ | $0.105 \pm 0.030$ | $0.00190 \pm 0.00321$ | $0.74 \pm 0.35$ |
| car | $0.124 \pm 0.040$ | $0.00171 \pm 0.00617$ | $0.83 \pm 0.28$ | $0.112 \pm 0.030$ | $0.00097 \pm 0.00187$ | $0.90 \pm 0.23$ |
| bird | $0.114 \pm 0.035$ | $0.00237 \pm 0.00425$ | $0.60 \pm 0.42$ | $0.097 \pm 0.033$ | $0.00140 \pm 0.00229$ | $0.84 \pm 0.28$ |
| cat | $\underline{0.093 \pm 0.031}$ | $0.00049 \pm 0.00093$ | $0.92 \pm 0.21$ | $0.095 \pm 0.026$ | $0.00043 \pm 0.00121$ | $0.99 \pm 0.05$ |
| deer | $0.123 \pm 0.042$ | $0.00221 \pm 0.00494$ | $0.86 \pm 0.26$ | $0.092 \pm 0.034$ | $0.00117 \pm 0.00180$ | $0.99 \pm 0.07$ |
| dog | $0.115 \pm 0.038$ | $\underline{0.00095 \pm 0.00128}$ | $0.91 \pm 0.19$ | $0.102 \pm 0.033$ | $0.00246 \pm 0.00526$ | $0.98 \pm 0.07$ |
| frog | $0.137 \pm 0.042$ | $0.00197 \pm 0.00219$ | $0.83 \pm 0.26$ | $0.112 \pm 0.035$ | $0.00158 \pm 0.00240$ | $0.98 \pm 0.07$ |
| horse | $\underline{0.106 \pm 0.041}$ | $\underline{0.00081 \pm 0.00227}$ | $0.69 \pm 0.36$ | $0.137 \pm 0.048$ | $0.00123 \pm 0.00214$ | $0.87 \pm 0.24$ |
| ship | $0.119 \pm 0.037$ | $0.00336 \pm 0.00497$ | $0.67 \pm 0.38$ | $0.085 \pm 0.031$ | $0.00232 \pm 0.00564$ | $0.93 \pm 0.19$ |
| truck | $0.125 \pm 0.033$ | $0.00130 \pm 0.00138$ | $0.80 \pm 0.28$ | $0.106 \pm 0.031$ | $0.00058 \pm 0.00097$ | $0.95 \pm 0.11$ |

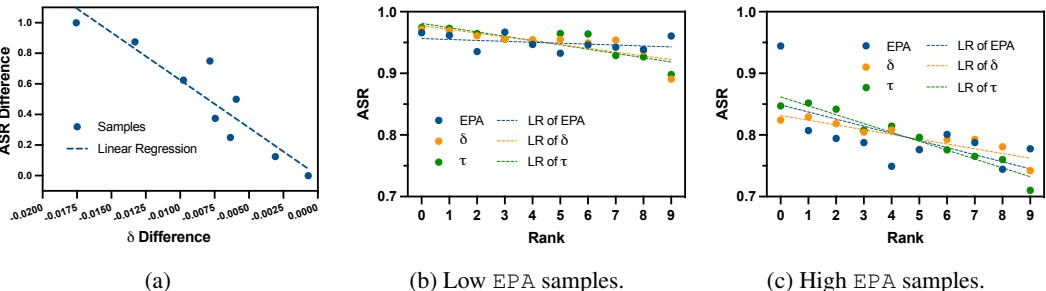

| (a) | (b) Low EPA samples. | (c) High EPA samples. |

Figure 5: (a) Correlation between pairwise $\delta$ difference and ASR difference; (b) and (c) Comparison between all of our metrics for low/high EPA samples.

### 4.3 POISONING DISTANCE AND POISONING BUDGET

In previous sections, we showed that EPA is generally a reliable indicator of poisoning difficulty. However, it has 2 major shortcomings: (1) it does not take into account $y_p$ in determining poisoning difficulty, and (2) it becomes ineffective within groups of target samples that have similar EPA. Motivated by the inadequacy of EPA, we apply the poisoning distance and the poison budget measure $\tau$, where our experience suggests that a larger $\delta$ or a larger $\tau$ indicates a more difficult attack (lower ASR). Specifically, given a target sample $\mathbf{x}_t$, we would like to confirm whether $\delta$ and $\tau$ are capable of predicting its vulnerability to poisoning towards a poison class $y_p$.

Specifically, we examine our prior results in the from-scratch setting. First, we look at the choice of $y_p$ with respect to each target class. For each target class, we examine the same 100 test samples (50 highest/lowest EPA targets) and calculate the average ASR for each $y_p$. We report the two $y_p$ classes with the lowest/highest average ASR and compare the average $\delta$ and $\tau$ values in Table 3. Note that we focus on *hard-to-classify* samples by performing a pre-screening process to rule out $(\mathbf{x}_t, y_p)$ pairs (1135 out of 9000) that are already classified as $y_p$ at the final epoch of clean training in any of the $M$ trials. While we observe $\delta$ and $\tau$ are generally capable of identifying easy/hard poison classes, there are some anomalies: for example, the target classes dog and cat have a smaller difference in ASR between the highest/lowest ASR $y_p$, making them more difficult to differentiate.

To provide further understanding on the role of the poison class $y_p$, for every individual instance $\mathbf{x}_t$, we enumerate the choice of $y_p$ and create pairs $((\mathbf{x}_t, y_p^1), (\mathbf{x}_t, y_p^2))$ (there are 9 choose 2, which is 36 pairs in total). For each pair, we calculate its corresponding ASR difference and $\delta$ difference. After obtaining all $1000 \times 36$ pairwise ASR and $\delta$ differences, we plot the correlation of the average $\delta$ difference with respect to the 9 possible ASR differences[9] in Figure 5(a) and observe that $\delta$ is generally reliable even for differentiating pair-wise differences.

---

[9] Note that for each attack instance, as we perform 8 trials, ASR can take 9 values in $[0, 1]$ with an interval of $1/8$. The ASR difference can only take the same 9 values as we restrict the difference to be positive.

Moreover, we previously mentioned that EPA is not very reliable in further identifying hard/easy to poison samples within the groups of samples with similar EPA. In Figure 5(b)(c), we rank samples in the high EPA and low EPA region according to their EPA/$\delta$/$\tau$ into 10 tiers and plot the average ASR for each tier. We observe that $\delta$ and $\tau$ cover a much wider range of ASR and are able to further predict the difficulty of poisoning in a more fine-grained way.

## 4.4 ABLATION STUDIES

**Necessity of training dynamics:** A simple baseline method to predict poisoning difficulty is the confidence of the target label $y_t$. We run GM on CIFAR-10 and set $y_t =$ plane, $y_p =$ bird and report the ASR, EPA, and confidence of $y_t$ for high EPA test samples and lower EPA samples[10] in Table 4. We observe that the confidence of $y_t$ is unable to differentiate samples that are difficult/easy to poison and it is necessary to consider the training dynamics with EPA.

Table 4: Ablation study on predicting poisoning difficulty with the confidence of $y_t$.

| $\mathbf{x}_t$ | confidence of $y_t$ | average EPA | average ASR |
|---|---|---|---|
| high EPA | $0.9985 \pm 0.00309$ | $0.9955 \pm 0.00165$ | $0.468 \pm 0.3718$ |
| lower EPA | $0.9999 \pm 0.00003$ | $0.9249 \pm 0.03210$ | $0.905 \pm 0.1899$ |

**Poison budget:** In our CIFAR-10 experiments using GM, the class "dog" is generally easier to attack, indicating the attack budget may be too high. While in such cases our EPA does not provide clear guidance, we conduct additional experiments by lowering the attack budget $\varepsilon$ in Table 5. We observe EPA is more reliable when the attacker's budget is limited, which is usually true in practice.

Table 5: Ablation study on predicting poisoning difficulty with EPA for various attack budget $\varepsilon$.

| $\mathbf{x}_t$ | $\varepsilon = 1\%$ | $\varepsilon = 0.75\%$ | $\varepsilon = 0.5\%$ | $\varepsilon = 0.25\%$ | $\varepsilon = 0.1\%$ |
|---|---|---|---|---|---|
| high EPA | $0.963 \pm 0.084$ | $0.963 \pm 0.119$ | $0.738 \pm 0.375$ | $0.588 \pm 0.382$ | $0.263 \pm 0.216$ |
| low EPA | $0.988 \pm 0.040$ | $1.000 \pm 0.000$ | $0.963 \pm 0.060$ | $0.950 \pm 0.121$ | $0.662 \pm 0.391$ |

**TinyImageNet**: Finally, we present our results on using EPA to predict poisoning difficulty on TinyImageNet in Table 6. We choose 4 highest/lowest samples from $y_t = 0$ and use GM and BP to poison towards $y_p = 1, 2$, with an attack budget $\varepsilon = 0.05\%$. Our results again confirm that EPA can predict poisoning difficulty.

Table 6: Poisoning difficulty prediction using EPA on TinyImageNet.

| $x_t$ | $y_p = 1$ | | $y_p = 2$ | |
|---|---|---|---|---|
| | GM | BP | GM | BP |
| high EPA | $0.188 \pm 0.375$ | $0.031 \pm 0.062$ | $0.531 \pm 0.373$ | $0.344 \pm 0.472$ |
| low EPA | $0.812 \pm 0.298$ | $0.781 \pm 0.438$ | $1.000 \pm 0.000$ | $0.781 \pm 0.438$ |

## 5 CONCLUSION

In this paper, we investigated the varying vulnerability of test samples to targeted data poisoning attacks. Our work establishes that poisoning difficulty is not uniform across samples but rather depends on specific, measurable characteristics. We identified three key factors, classification difficulty during clean training, parameter space poisoning distance, and poison budget, that significantly influence vulnerability. Based on these factors, we developed three complementary metrics: ergodic prediction accuracy (EPA), poisoning distance and budget lower bound. These metrics provide a comprehensive framework for predicting poisoning difficulty without requiring the execution of actual attacks. Our experimental results validate that EPA effectively separates easy-to-poison from hard-to-poison samples, while $\delta$ and $\tau$ offer fine-grained predictions for specific poison classes. We extend more discussions on limitations and future works in Appendix A.

---

[10]We restrict the confidence of $y_t > 0.98$ for samples with lower EPA.

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

## A    LIMITATIONS AND FUTURE WORKS

**Practical Utility for Defenders.**    The framework proposed in this paper has potential for practical application in defending against data poisoning attacks. Defenders could integrate these metrics into a continuous monitoring pipeline to achieve proactive vulnerability assessment. For instance, `EPA` could be periodically computed for a set of critical or high-stakes test samples to identify any that exhibit unstable predictions during routine model updates. For samples flagged with low `EPA`, or for particularly sensitive targets (e.g., a specific face in a facial recognition system), a defender could then compute $\delta$ and $\tau$ against a set of likely or dangerous poison classes. A high-risk combination could trigger an alert, mandate human-in-the-loop verification for that sample's predictions, or initiate a forensic analysis of the training data pipeline.

**Limitations.**    Our work still has limitations: (1) As shown in our experiments, all of our metrics may occasionally yield inaccurate predictions of poisoning difficulty, indicating room for improvement; (2) Our method still relies on test sample labels to accurately predict poisoning difficulty, which could be infeasible in practice. Thus, a label-agnostic approach is largely desired; (3)Our quantitative metrics are currently limited to classification models, with our diffusion model analysis in Appendix E remaining qualitative.

**Future Works.**    Several potential future directions emerge from our work: (1) Data-centric defenses that optimize test samples to defend against targeted data poisoning attacks. For example, defenders might apply carefully crafted adversarial noise to test data, similar to techniques used in adversarial examples; (2) While we make initial attempts to extend our discussion to diffusion models in Appendix E, future work could explore how these vulnerability insights inform the development of more robust LLMs against targeted data poisoning attacks. Moreover, developing universal quantitative metrics for assessing poisoning difficulty across different model types is a crucial future step.

## B    RELATED WORKS

### B.1    DATA POISONING ATTACKS

Data poisoning, an emerging training-time concern in modern ML pipelines, refers to the threat of (actively or passively) crafting "poisoned" training data $\mathcal{D}_p$ so that systems trained on it (along with possibly clean in-house data $\mathcal{D}_c$) are skewed toward certain behaviors. Significant research has been proposed to study the impact of such attacks on classification models. For example, indiscriminate data poisoning (e.g., Biggio et al., 2012; Koh & Liang, 2017; Koh et al., 2022; Muñoz-González et al., 2017; Lu et al., 2022; 2023; 2024a;b;c) is a general-purpose attack that aims to decrease the overall test accuracy. Similar formulations have been proposed for protecting user data (e.g., Liu & Chawla, 2010; Huang et al., 2021; Yu et al., 2022; Fowl et al., 2021b;a; Sandoval-Segura et al., 2022; Fu et al., 2021). While data poisoning attacks can also involve testing-time manipulation—such as backdoor attacks (e.g., Gu et al., 2017; Tran et al., 2018; Chen et al., 2017; Saha et al., 2020) that aim to trigger malicious model behavior with particular patterns on test samples, we focus exclusively on training-time attacks in this paper.

### B.2    ADDING ATTACK VS REPLACING ATTACK

Realistically, an attacker would have no control on the clean set $\mathcal{D}_c$, and data poisoning attacks (e.g., Biggio et al., 2012; Koh et al., 2022; Muñoz-González et al., 2017; Lu et al., 2022; 2023) usually consider *adding-only* attack where $\mathcal{D}_c$ is intact and the size of $\mathcal{D}_{tr}$ increases. However, targeted attacks (e.g., Shafahi et al., 2018; Aghakhani et al., 2021; Guo & Liu, 2020; Zhu et al., 2019) consider *replacing* attacks where part of the clean set $\mathcal{D}_c$ is substituted with $\mathcal{D}_p$ while the size of $\mathcal{D}_{tr}$ is unchanged. Note that such a substitution is performed by simply adding noise to the original clean samples. Such a setting could resonate in targeted settings as it would keep the balance between classes. In this paper, we follow previous works and consider *replacing attacks*. While the practicality of such attacks are beyond our scope, we note that the key technical differences comparing with adding-only attacks: replacing attacks are notably easier as it reduces $|\mathcal{D}_c|$ and

considers a slightly higher $\varepsilon$ as $|\mathcal{D}_{tr}|$ is a constant (see Appendix C.10 in (Lu et al., 2023) for a detailed discussion).

### B.3 MEASURING CLASSIFICATION DIFFICULTY

The problem of measuring classification difficulty has been explored in prior literature. For instance, Agarwal et al. (2022) proposed variance of gradient (VOG) as a method to rank examples by classification difficulty. VOG could potentially serve as an alternative to `EPA` for measuring classification difficulty in Hypothesis A and may function as an indicator for predicting poisoning vulnerability—a direction we intend to investigate in future work. Additionally, out-of-distribution (OoD) detection techniques such as PCA (Hawkins, 1974) and KDE (Davis et al., 2011) could potentially identify *hard-to-classify* (and possibly *easy-to-poison*) anomalous samples.

Furthermore, our approach relates to selective classification Rabanser et al. (2022), where models reject inputs likely to be misclassified while maintaining high performance on accepted inputs. Specifically, Rabanser et al. (2022) leverages prediction agreement between intermediate training stages and the final epoch—a strategy similar to our `EPA` metric that also analyzes clean training dynamics. However, unlike selective classification, we assign a `EPA` score to every test instance rather than implementing a rejection mechanism.

### B.4 CONNECTION AND DIFFERENCES WITH MODEL-TARGETED ATTACKS

We note that our core idea of poisoning distance is closely related to model-targeted indiscriminate attacks which we denote as MTA (Suya et al., 2021; Lu et al., 2023; Koh et al., 2022), where these attacks consider a set of target parameters $\mathbf{w}_p$ as the target and apply gradient-based poisoning attacks to achieve $\mathbf{w}_p$. While the concept of target parameters is also used in our paper, we emphasize key differences: (1) *Task*: MTA considers $\mathbf{w}_p$ to be a model with low test accuracy, which can be generated with a gradient-based parameter corruption attack (Sun et al., 2020). We consider a set of $\mathbf{w}_p$ that only misclassifies one single test sample. (2) *Using* $\mathbf{w}_p$: MTA uses $\mathbf{w}_p$ as the endgoal to generate poisoning attacks, we use $\mathbf{w}_p$ as proxies to quantify poisoning difficulty. (3) *Attack vs Defense*: MTA are designed for more effective attacks, while our algorithm estimates $\mathbf{w}_p$ and $\delta$ to help practitioners understand targeted attack difficulties and design better defenses.

---

**Algorithm 1:** Poisoning Distance Estimation

**Input:** clean parameters $\mathbf{w}_c$, target $\mathbf{x}_t$, poison label $y_p$, precision parameter $\alpha = 10^{-4}$

1   calculate the gradient $\mathbf{g} = \nabla_{\mathbf{w}_c} \ell(f(\mathbf{x}_t; \mathbf{w}_c), y_p)$
2   instantiate the upper bound $u = \infty$, lower bound $l = 0$, and medium $m = 0.5$ for binary search
3   **while** $u - l > \alpha$ **do**
4      **if** $u = \infty$ **then**
5         set $m = 2m$
6      **else**
7         set $m = \frac{u+l}{2}$
8      **if** $f(\mathbf{x}_t; \mathbf{w} - m \cdot \mathbf{g}) = y_p$ **then**
9         set $u = m$
10     **else**
11        set $l = m$
12   **return** the estimated poisoning distance $\delta = u$

---

## C   ALGORITHM ON ESTIMATING POISONING DISTANCE

Recall that in Section 3.2 we propose to use a binary search based algorithm to estimate the poisoning distance $\delta$. We present the algorithm in Algorithm 1.

# D  ADDITIONAL EXPERIMENTS ON CLASSIFICATION MODELS

## D.1  COMPUTING RESOURCE & TIME

**Targeted attacks:** Due to the extensive number of attacks conducted, we distributed our experiments across three distinct clusters equipped with NVIDIA 4090 (cluster 1), A100 (cluster 2), and RTX6000 GPUs (cluster 3). The computational requirements varied significantly by task: training models from scratch (GM experiments) required up to 1 hour 40 minutes on all clusters for CIFAR-10/ResNet-18 configurations, while TinyImageNet/VGG16 experiments (GM and BP) demanded up to 3 hours 10 minutes on cluster 2. Transfer learning experiments were considerably more efficient, requiring only 66-72 seconds for BP and 60-63 seconds for FC on clusters 2 and 3, respectively.

**Measuring poisoning difficulty:** We conducted all experiments on the NVIDIA 4090 cluster. For EPA calculations, the computational cost scales linearly with the number of trials $M$ multiplied by the clean training time. For individual test samples, computing all nine possible $\delta$ values for a single set of $\mathbf{w}_c$ requires just 1.3 seconds, while calculating all nine possible $\tau$ values takes approximately 30 seconds on our ResNet-18/CIFAR-10 experimental setup.

## D.2  OTHER BASELINES FOR POISONING DIFFICULTY PREDICTION ON $y_p$

In Section 4.3, we introduce $\delta$ as the indicator for poisoning difficulty for poison classes $y_p$ with the highest and lowest ASR. Here we perform the same task for two baseline methods: (1) the average confidence of a given $y_p$ at the end of clean training; (2) the poison prediction area (PPA), where we apply a similar definition with EPA, but only considers the prediction of $y_p$. Our results in Table 7 and Table 8 show that these baseline methods fail to predict poisoning difficulty for most cases.

Table 7: Measuring the poisoning difficulty of GM on CIFAR-10 using the average confidence of $y_p$ at the end of clean training with the highest and lowest ASR over all target classes $y_t$. Cases where the prediction conforms with ASR are marked in blue, and anomalies are marked in red.

| $y_t$ | Lowest ASR $y_p$ | | Highest ASR $y_p$ | |
|---|---|---|---|---|
| | Avg Confidence of $y_p$ | ASR | Avg Confidence of $y_p$ | ASR |
| plane | $0.002 \pm 0.005$ | $0.57 \pm 0.42$ | $0.001 \pm 0.005$ | $0.74 \pm 0.35$ |
| car | $0.001 \pm 0.001$ | $0.83 \pm 0.28$ | $0.000 \pm 0.001$ | $0.90 \pm 0.23$ |
| bird | $0.001 \pm 0.003$ | $0.60 \pm 0.42$ | $0.003 \pm 0.008$ | $0.84 \pm 0.28$ |
| cat | $0.002 \pm 0.006$ | $0.92 \pm 0.21$ | $0.001 \pm 0.004$ | $0.99 \pm 0.05$ |
| deer | $0.002 \pm 0.006$ | $0.86 \pm 0.26$ | $0.003 \pm 0.006$ | $0.99 \pm 0.07$ |
| dog | $0.002 \pm 0.005$ | $0.91 \pm 0.19$ | $0.002 \pm 0.006$ | $0.98 \pm 0.07$ |
| frog | $0.002 \pm 0.005$ | $0.83 \pm 0.26$ | $0.001 \pm 0.004$ | $0.98 \pm 0.07$ |
| horse | $0.003 \pm 0.011$ | $0.69 \pm 0.36$ | $0.002 \pm 0.005$ | $0.87 \pm 0.24$ |
| ship | $0.001 \pm 0.002$ | $0.67 \pm 0.38$ | $0.002 \pm 0.006$ | $0.93 \pm 0.19$ |
| truck | $0.000 \pm 0.002$ | $0.80 \pm 0.28$ | $0.001 \pm 0.004$ | $0.95 \pm 0.11$ |

# E  TARGETED ATTACKS ON LATENT DIFFUSION MODELS

For generative models, targeted attacks aim to alter the generation of specific concepts or prompts while maintaining expected behavior for other inputs (Lu et al., 2024c; Shan et al., 2024; Wan et al., 2023). In this paper, we aim to extend the examination on instance-level difficulty in targeted data poisoning attacks to latent diffusion models Rombach et al. (2022). Specifically, we consider the disguised copyright infringement (DCI) attack in Lu et al. (2024c). DCI considers a threat where an attacker aims to mimic the style of a copyrighted image $x_c$ without directly training on $x_c$, but creates a disguise image (or poison sample) $x_d$ that visually assembles another base image $x_b$ while containing the latent information of $x_c$. Although the task and poisoning mechanism are very different from those of classification models that we consider in the main paper, we find two factors that would affect the poisoning difficulty.

Table 8: Measuring the poisoning difficulty of GM on CIFAR-10 using poison prediction area (PPA) with the highest and lowest ASR over all target classes $y_t$. Cases where the prediction conforms with ASR are marked with blue, and anomalies are marked with red.

| $y_t$ | Lowest ASR $y_p$ | | Highest ASR $y_p$ | |
|---|---|---|---|---|
| | PPA | ASR | PPA | ASR |
| plane | $0.004 \pm 0.016$ | $0.57 \pm 0.42$ | $0.003 \pm 0.012$ | $0.74 \pm 0.35$ |
| car | $0.004 \pm 0.014$ | $0.83 \pm 0.28$ | $0.001 \pm 0.005$ | $0.90 \pm 0.23$ |
| bird | $0.005 \pm 0.009$ | $0.60 \pm 0.42$ | $0.005 \pm 0.010$ | $0.84 \pm 0.28$ |
| cat | $0.016 \pm 0.026$ | $0.92 \pm 0.21$ | $0.006 \pm 0.013$ | $0.99 \pm 0.05$ |
| deer | $0.017 \pm 0.031$ | $0.86 \pm 0.26$ | $0.004 \pm 0.007$ | $0.99 \pm 0.07$ |
| dog | $0.008 \pm 0.013$ | $0.91 \pm 0.19$ | $0.003 \pm 0.011$ | $0.98 \pm 0.07$ |
| frog | $0.003 \pm 0.012$ | $0.83 \pm 0.26$ | $0.006 \pm 0.015$ | $0.98 \pm 0.07$ |
| horse | $0.012 \pm 0.042$ | $0.69 \pm 0.36$ | $0.003 \pm 0.013$ | $0.87 \pm 0.24$ |
| ship | $0.004 \pm 0.015$ | $0.67 \pm 0.38$ | $0.004 \pm 0.012$ | $0.93 \pm 0.19$ |
| truck | $0.002 \pm 0.008$ | $0.80 \pm 0.28$ | $0.002 \pm 0.007$ | $0.95 \pm 0.11$ |

**Structure of $x_b$:** We follow the implementation of Lu et al. (2024c)[11] and consider the task of disguising style. We pick the drawing: The Neckarfront in Tubingen, Germany (photo by Andreas Praefcke) in the style of *The Starry Night*, generated with Neural Style Transfer (Gatys et al., 2015) as $x_c$. The base image $x_b$ is $x_c$ with another style (watercolor), generated with AdaIN-based (Huang & Belongie, 2017) style transfer[12]. The disguise $x_d$ is generated using Algorithm 1 in Lu et al. (2024c) and we train the disguise $x_d$ using textual inversion Gal et al. (2023) for generation. We fix $x_t$ and study the role of the structure of $x_b$ by applying gaussian blur with different kernel size (a larger kernel size results in a more blurry image). We report our results in Figure 6, Figure 7, Figure 8, Figure 9 and Figure 10. We observe that by increasing the kernel size, the cirrus effect of the generated images dramatically decreases. When the kernel size is bigger than 10, the textual inversion model cannot learn any useful information. We conclude that preserving the structure of $x_b$ is essential for a successful data poisoning attack, highlighting the role on the appearance of the poison image in poisoning difficulty.

**Structure of $x_c$:** We also observe that the structure of $x_c$ (target or copyright image) also affects the poisoning difficulty. In Figure 11, we choose another $x_t$[13] with a much simpler layout in the same style as *The Starry Night*. We observe that style mimicry is unsuccessful for this poison instance, validating that poisoning success is also correlated with the structure of $x_c$.

---

[11] https://github.com/watml/disguised_copyright_infringement

[12] https://github.com/tyui592/AdaIN_Pytorch

[13] https://stock.adobe.com/images/glowing-moon-on-a-blue-sky-abstract-background-seamless-vector-pattern-in-the-style-of-impressionist-paintings/475101004

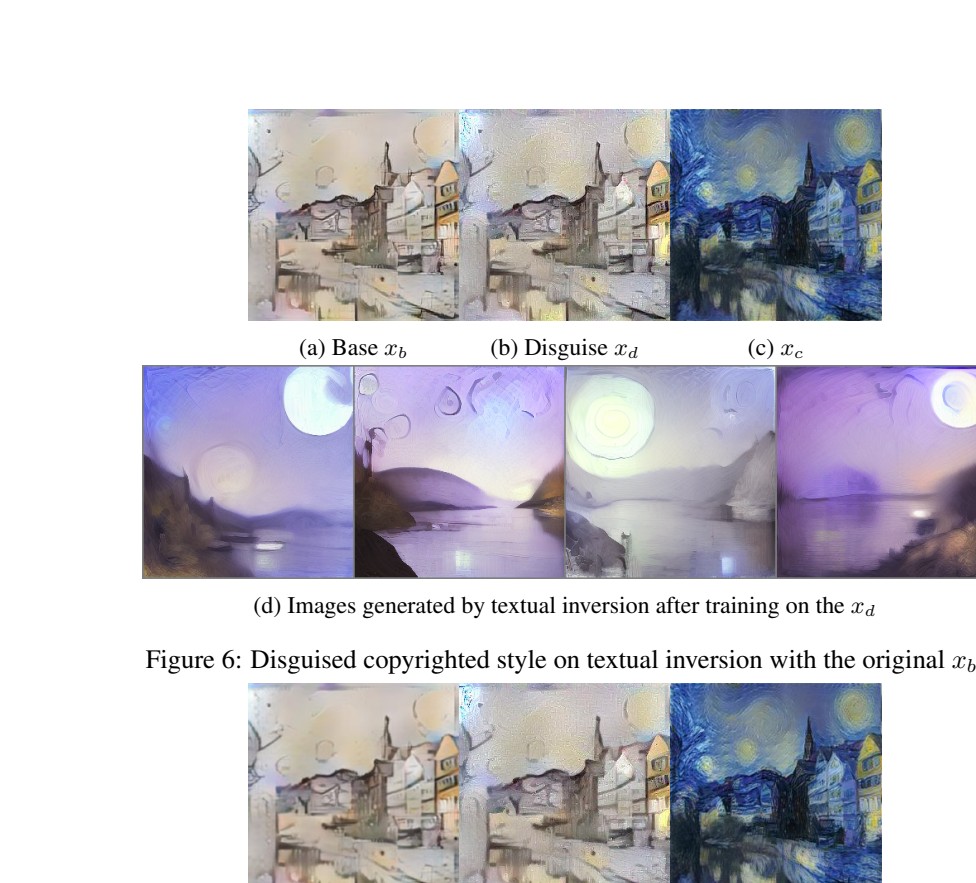

(a) Base $x_b$      (b) Disguise $x_d$      (c) $x_c$

(d) Images generated by textual inversion after training on the $x_d$

Figure 6: Disguised copyrighted style on textual inversion with the original $x_b$.

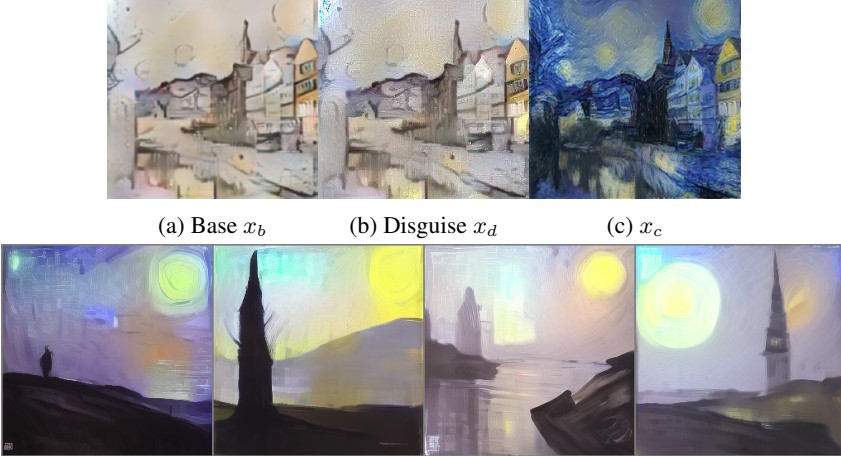

(a) Base $x_b$      (b) Disguise $x_d$      (c) $x_c$

(d) Images generated by textual inversion after training on the $x_d$

Figure 7: Disguised copyrighted style on textual inversion with the blurry $x_b$ (kernel size = 3).

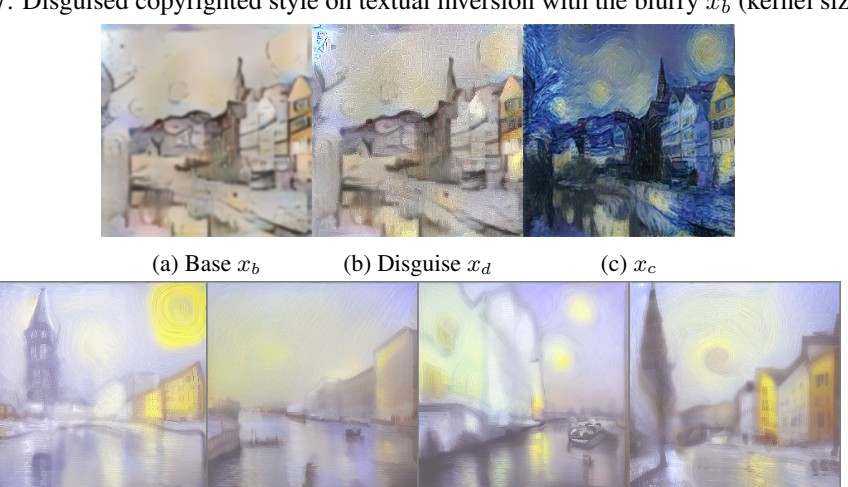

(a) Base $x_b$      (b) Disguise $x_d$      (c) $x_c$

(d) Images generated by textual inversion after training on the $x_d$

Figure 8: Disguised copyrighted style on textual inversion with the blurry $x_b$ (kernel size = 7).

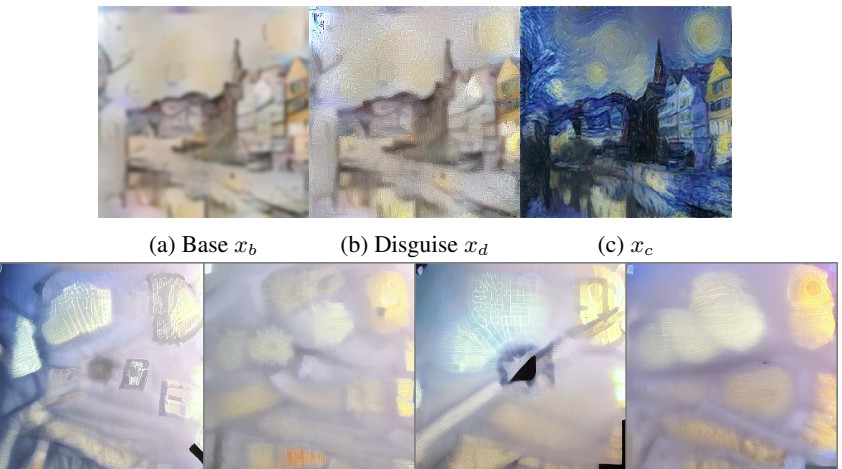

(a) Base $x_b$     (b) Disguise $x_d$     (c) $x_c$

(d) Images generated by textual inversion after training on the $x_d$

Figure 9: Disguised copyrighted style on textual inversion with the blurry $x_b$ (kernel size = 13).

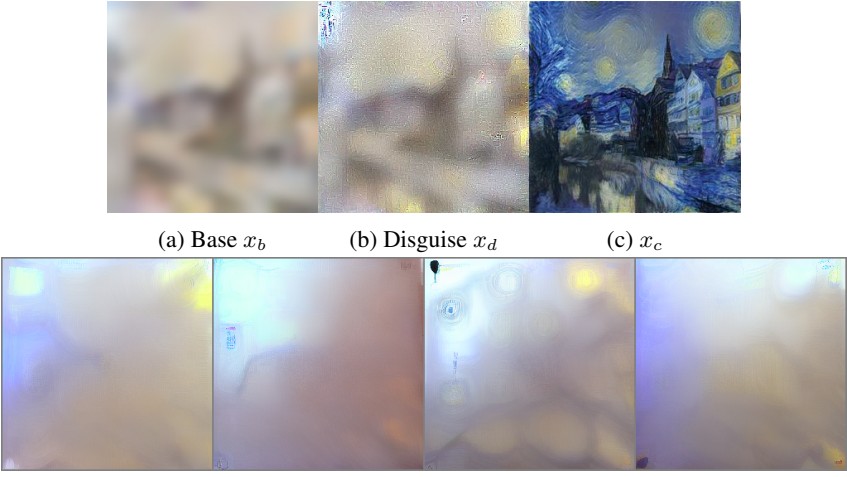

(a) Base $x_b$     (b) Disguise $x_d$     (c) $x_c$

(d) Images generated by textual inversion after training on the $x_d$

Figure 10: Disguised copyrighted style on textual inversion with the blurry $x_b$ (kernel size = 49).

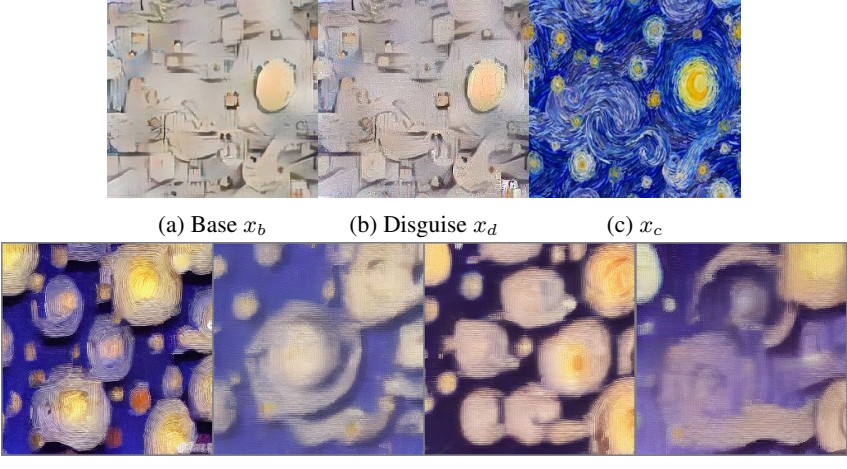

(a) Base $x_b$     (b) Disguise $x_d$     (c) $x_c$

(d) Images generated by textual inversion after training on the $x_d$

Figure 11: Disguised copyrighted style on textual inversion with a different choice of $x_t$.

