# OpenReview forum: "Not All Samples Are Equal: Quantifying Instance-level Difficulty in Targeted Data Poisoning"
_ICLR.cc/2026/Conference — Submitted to ICLR 2026_

### Official Review · Reviewer_cVWR · 2025-10-28

**Soundness:** 3
**Presentation:** 3
**Contribution:** 3
**Rating:** 6
**Confidence:** 3

**Summary:**

The paper studies instance-level difficulty for targeted data poisoning. Instead of reporting averaged attack success rates (ASR), it asks why some test samples are easier to poison than others and proposes three predictive metrics computed without running an attack: (i) Ergodic Prediction Accuracy (EPA) from clean-training dynamics; (ii) a poisoning distance δ, the minimal one-step parameter move along the sample’s gradient that flips the target sample to a chosen poison class; and (iii) a lower bound on poison budget τ derived from a reachability condition (Lambert-W expression based on Lu et al., 2023). Across CIFAR-10 (ResNet-18) and a TinyImageNet ablation (VGG-16) with GM/FC/BP attacks, the authors show: EPA separates easy vs. hard targets; δ and τ provide finer ranking conditioned on poison class; and results hold under different budgets and training regimes (from-scratch vs. linear-head transfer). The work argues these metrics can guide defenders to pre-screen vulnerable instances.

**Strengths:**

- The paper asks a concrete, useful question at the right granularity and sticks to it.

- Experiments are careful and show stable trends across setups rather than one-off wins.

- The ideas are easy to plug into existing workflows and yield actionable guidance for practitioners.

**Weaknesses:**

- Scalability is questionable since the training-dynamics signal is computationally expensive at realistic scales.
- The metrics are proxies rather than outcomes of an actual poisoning process, and robustness across optimizers or norms is underexplored.
- Evidence is concentrated on standard image classification; generalization to larger models, other modalities, or backdoor threats remains unclear.
- The paper does not report calibrated predictive utility (for example AUC or operational thresholds), which limits direct deployment guidance.

**Questions:**

See weakness as above.

---

> ### Author Response · Authors · 2025-11-18
>
> We thank the reviewer for the positive feedback and insightful comments, and we address your concerns below:
>
> **[W1: Scalability]**
> - We agree that EPA could add an extra cost to training. In practical implementations, a proxy EPA can be performed by restricting to high-stakes test samples and limiting its frequency (e.g., computed at every 10 epochs), as listed in line 707 in Appendix A. Straightforward parallelism (over initializer and samples) can also be exploited to reduce the cost.
> - We believe it is important to understand poisoning difficulty as a fundamental scientific question, and the extra cost is worthwhile given what could be at stake.
>
> **[W2: Actual poison]** Our metrics are intentionally designed to avoid considering any concrete attacks for generalization. Instead, we aim to predict the vulnerability of a test sample against the *most powerful attacks*.
> - As a result, we deploy attacks with hyperparameters that represent the best-performing attacks. Tuning such hyperparameters would lead to suboptimal attacks with lower ASR but does not affect our prediction tools.
>
> **[W3: Other models and attacks]**
> - Indeed we primarily focus on targeted data poisoning attacks on image classification models, where gradient-based attacks (such as gradient matching) are performed. Such attacks (e.g., Geiping et al., 2021) are conducted in the setting of CIFAR-10/TinyImageNet with ResNet-18/VGG-16, and we directly adopt their settings.
> - While we appreciate the reviewer's suggestions on multimodal systems, we are not aware of existing gradient-based targeted attacks on such models (as text poisoning using gradients is difficult).
> - Moreover, we believe that even in our setting, poisoning attack difficulty is poorly understood, and we believe such a study is valuable.
> - We would like to draw your attention to our Appendix E, where we consider targeted attacks on generative models. However, as the threat model is vastly different, our proposed tools do not directly transfer.
> - While we agree that backdoor attacks are interesting, they consider a very different threat model where a trigger is also injected at test time. As a result, poisoning difficulty would be largely associated with the trigger, and our results would not be readily applicable.
>
> **[W4: Calibrated predictive utility]** We have performed an experiment on calibrated predictive utility to predict the success of targeted data poisoning attacks (GM) on CIFAR-10. We observe an AUC (area under the curve) of 0.93 without specifying the poisoning label $y_p$​ (using only EPA) and accuracies of 87.5% and 88.2% for identifying the poisoned $y_p$​ using $\delta$ and $\tau$, respectively.

---

> > ### Comment · Reviewer_cVWR · 2025-11-27
> >
> > Thanks for the detailed responses. The responses address my concerns. Thus I raise my rating to an 8.

---

> > > ### Author Response · Authors · 2025-11-28
> > >
> > > We are glad that your concerns have been addressed and thank you for recognizing our contribution! We really appreciate your effort and insights in helping us improve the paper.

---

### Official Review · Reviewer_EM8p · 2025-10-31

**Soundness:** 2
**Presentation:** 3
**Contribution:** 2
**Rating:** 4
**Confidence:** 4

**Summary:**

In this paper, the authors studied data poisoning attacks for image classification. In particular, what are the main factors that affect the success rate of a targeted poisoning attack. They have identified three factors: 1) classification difficulty for the clean model, 2) poisoning distance and 3) poison budget, as the main factors. They provide some insights about why these three factors may affect the attack successful rate and also how to compute those factors with access to only the clean model, training data and the targeted test sample. They have demonstrated the correlation between these factors and the ASR on attacking models trained on CIFAR-10 and Tiny-ImageNet.

**Strengths:**

The authors studied an interesting problem and the structure of the paper is clear and easy to follow.

**Weaknesses:**

I listed a few weaknesses on the paper as follows:
1. It is not very clear how the authors measure and determine if the attack is success or not. There are at least two criteria to evaluate the success of a poisoning attack at test time: 1) if the test sample $x_t$ is successfully mis-classified as the target class $y_p$; 2) if the overall model performance on other test samples are almost the same as a clean model. I do not see any mentioned of the 2nd criteria in the discussion.
2. Are there any co-relation among these three metrics? Does one affect the other two? As computing them require different information and compute resources (one may be more costly than the other), does one need to compute all of them to understand attack difficulty?
3. Is there any other factor that may affect the attack difficulty? Like the hyperparameters setting of the attack algorithm, the model architecture? More ablation study should be conducted to evaluate these factors.

**Questions:**

1. In figure 2, it seems like in general the ASRs are higher for high EPA groups than low EPA groups per poison class, however, the variance of the high EPA groups are much larger than low EPA groups. It makes it hard to tell if EPA is a good metric to determine attach difficulty? There is no explanation on why the variance for high EPA groups is almost twice larger than low EPA groups.
2. The 3rd metric is already proposed by Lu et al. (2023)? So this is not a new metric?
3. Fig 3 a) looks different than Figure 2, what have changed in the experimental setting?
4. In the paragraph discuss Fig 3, it seems like per class accuracy is also a very important indicator for attack difficulty? How important is per class accuracy comparing to EPA?

---

> ### Author Response · Authors · 2025-11-18
>
> We thank the reviewer for the insightful feedback, and we address your concerns below:
>
> **[W1: Measuring attack success]** The reviewer is right that the overall model performance on other test samples should also be checked. Indeed, we have compared the overall test accuracy of the poisoned model with its clean counterparts across all attacks and models, and observed that the difference is always less than 1%, confirming the second criterion.
>
> **[W2: Co-relation]**
> - The poisoning distance and poison budget are indeed interconnected, as they share the same proxy of $\mathbf{w}_p$​ (the target parameter) and require similar information and computation. However, they differ in two important respects: $\delta$ directly measures the model distance, while $\tau$ measures the minimum poisoning budget required to achieve $\mathbf{w}_p$​.
> - In Appendix A (Practical Utility for Defenders) we extend the discussion on how to use the three metrics. We divide the calculation of EPA and the other two metrics into two steps: (1) EPA can roughly determine which $x_t$ is more vulnerable, (2) while $\delta$ and $\tau$ provide a fine-grained prediction on the poison classes.
>
>
> **[W3: Other factor]**
> - We would like to clarify that our three metrics do not depend on any specific attack and we aim to predict the vulnerability of a test sample against the most powerful attacks.
> - As a result, we deploy attacks with hyperparameters that represent the best-performing attacks. Tuning such hyperparameters would lead to suboptimal attacks with lower ASR but does not affect our prediction tools.
>
> **[Q1: Figure 2]**
> - We would like to clarify that Figure 2 demonstrates the variance across different choices of the **poison class** $y_p$​ (the class that the attacker aims to misclassify to).
> - As a result, each bin in Figure 2 may contain samples from different target classes $y_t$​ and does not fully represent the performance of EPA. We refer the reviewer to Figure 3 for a more accurate presentation.
> - Across all experiments, we observe that test samples with low EPA are much more vulnerable and likely yield high ASR without large variance. However, for those with high EPA, ASR may be extremely low (e.g., 0%), thus causing the high variance.
>
> **[Q2: Lu et al.,]** Indeed, Theorem 1 is a direct adaptation from Lu et al., 2023. However, we want to emphasize that poisoning reachability was a tool proposed by Lu et al. to study indiscriminate attacks, and its application to predicting the poisoning difficulty of targeted attacks was underexplored.
>
> **[Q3: Figure 3(a)]**
> - We clarify that the x-axis of Figure 2 represents the poison class $y_p$​, while that of Figure 3(a) represents the target class $y_t$.
> - We recall that the target label $y_t$ is the actual label of $x_t$; the poisoning label $y_p$ is the label we want the poisoned model to predict as.
> - As a result, they serve different purposes: Figure 2 shows the variance between different choices of $y_p$ by the attacker (which motivates the other two metrics), while Figure 3(a) demonstrates the effectiveness of EPA without a fine-grained choice of $y_p$.
>
> **[Q4: Class accuracy]**
> - Indeed, class accuracy is a very coarse indicator and can be used to roughly determine samples from which target class is more likely to be poisoned. However, it cannot be used to determine sample-wise poisoning difficulty.
> - Class accuracy can be used to understand the variance returned by EPA, as presented in line 355.

---

> > ### Comment · Reviewer_EM8p · 2025-11-26
> >
> > Thanks for the detailed responses. The responses answered some of my questions. I have a few follow-up questions:
> > 1. Since "$\delta$ and $\tau$ provide a fine-grained prediction on the poison classes'', similarly, the class accuracy also tells if a class is easy to be manipulated, the questions is that do we need to spend those compute cost to make a very accurate prediction, or just use class accuracy? It is better to include such discussions.
> > 2. In terms of other factors, I want to understand how important is other factors vs. the three factors studied in the paper. It could be, for example, choosing hyperparameters or use a less-sensitive model (small models vs large models) will directly increase the difficulty of performing poisoning attacks. These factors should be discussed in the paper. Therefore, the analysis is more comprehensive.

---

> > > ### Author Response · Authors · 2025-11-26
> > >
> > > Thank you for the follow-up questions. Here we further address your concerns:
> > >
> > > **[Question 1]** We would like to clarify that $\delta$ and $\tau$ provide a fine-grained prediction of the **poison classes**, while class accuracy indicates which **target class samples** are (relatively) easy to poison. Thus, the latter does not provide any additional information beyond EPA, and the calculation of $\delta$ and $\tau$ is necessary to determine the difficulty of poisoning a target class to a specific poison class.
> > >
> > > To be more precise, consider the following example. For our test sample of interest $(x_t,y_t)$, class accuracy only tells us that samples in $y_t$​ may be easy or difficult to poison, while $\delta$ and $\tau$ precisely characterize how easy it is for an attacker to manipulate $y_t$ to $y_p$.
> > >
> > > **[Question 2]** We believe the other factors the reviewer mentioned relate to the **poisoning effectiveness** of a specific *attack algorithm* with respect to training hyperparameters, not **poisoning difficulty** of *test samples*. We clarify further below:
> > >
> > > - We agree that changing these factors would affect a specific attack algorithm. For example, using a different model than the one considered by the attack would result in a suboptimal attack (this is generally discussed as *transferability* in poisoning attack literature).
> > > - However, rather than considering one specific attack, our tool assesses poisoning difficulty against any data poisoning attack.
> > > - Furthermore, such training hyperparameters are captured by our EPA, which naturally considers poisoning difficulty in such settings.

---

> > > > ### Author Response · Authors · 2025-11-28
> > > >
> > > > Dear Reviewer EM8p,
> > > >
> > > > Thank you again for your prompt and thoughtful engagement in the discussion period.
> > > > As the discussion period draws to a close, we wanted to follow up to ensure your remaining concerns have been fully addressed. If any questions or uncertainties remain, we would be happy to provide further clarification.
> > > >
> > > > We believe our revisions and responses have meaningfully addressed the issues you raised. If you find our rebuttal satisfactory, we would greatly appreciate if you would consider adjusting your rating accordingly.
> > > > Thank you for your time and constructive engagement with our work.
> > > >
> > > > Best regards,
> > > >
> > > > Authors

---

### Official Review · Reviewer_t48Y · 2025-11-03

**Soundness:** 2
**Presentation:** 2
**Contribution:** 2
**Rating:** 4
**Confidence:** 3

**Summary:**

This paper investigates the heterogeneity in vulnerability among individual test instances under targeted data poisoning attacks. Unlike prior works that evaluate attack success rates in aggregate, the authors propose three predictive metrics: Ergodic Prediction Accuracy (EPA), Poisoning Distance, and Poison Budget Lower Bound, to quantify instance-level poisoning difficulty. These metrics are derived solely from clean training dynamics and model parameters, enabling defenders to assess vulnerability without performing actual attacks. The authors demonstrate strong correlations between these metrics and empirical attack success rates. The paper contributes a conceptual and empirical framework for understanding and predicting targeted poisoning susceptibility.

**Strengths:**

1. The work introduces a fresh viewpoint by shifting the analysis from aggregate attack success rates to per-sample poisoning difficulty, a dimension rarely explored in the literature on adversarial and poisoning attacks.
2. The three complementary metrics (EPA, $\delta$, $\tau$) cover distinct yet interrelated aspects of vulnerability: training dynamics, model-space perturbations, and resource constraints, forming a coherent conceptual framework.
3. The proposed metrics can be computed without executing any attack, making the approach practical for real-world defensive analysis where retraining and large-scale attack simulations are infeasible.

**Weaknesses:**

1. The theoretical foundation for metric relationships is limited. While there exits some empirical correlations, the paper lacks formal theoretical grounding explaining why EPA, $\delta$, and $\tau$ should predict poisoning difficulty beyond intuitive hypotheses. The metrics’ interdependence and causal relationships remain underexplored.
2. Experiments are primarily limited to CIFAR-10 and TinyImageNet with ResNet-18 and VGG-16. The findings may not generalize to large-scale or non-vision models such as multimodal systems, where training dynamics differ significantly.
3. The sensitivity and calibration of metrics is unclear. The quantitative scales of $\delta$ and $\tau$ lack interpretability, and there is no discussion on thresholds or normalization, making it difficult for practitioners to operationalize these measures in practice.
4. The analysis of failure cases and anomalies is incomplete. The authors acknowledge anomalies where $\delta$ or $\tau$ predictions deviate from observed attack success, but provide limited analysis of these inconsistencies, leaving open questions about robustness under different data distributions.

**Questions:**

Please refer to weaknesses.

---

> ### Author Response · Authors · 2025-11-18
>
> We thank the reviewer for the insightful feedback, and we address your concerns below:
>
> **[1. Theoretical foundation]** We appreciate the reviewer's insights on theoretical foundations and would like to emphasize the following:
> - For Ergodic Prediction Accuracy (EPA) and poisoning distance $\delta$, it is extremely difficult to derive theoretical guarantees without considering concrete data poisoning attacks. We believe our Hypotheses A and B are empirically validated by our experiments.
> - The poison budget $\tau$ does have a theoretical guarantee provided by Lu et al. (2023) in Theorem 1. However, we want to emphasize that poisoning reachability was a tool proposed by Lu et al. to study indiscriminate attacks, and its application to predicting the poisoning difficulty of targeted attacks was not explored before our work.
> - The poisoning distance and poison budget are indeed interconnected, as they share the same proxy of $\mathbf{w}_p$​ (the target parameter). However, they differ in two important respects: $\delta$ directly measures the model distance, while $\tau$ measures the minimum poisoning budget required to achieve $\mathbf{w}_p$​.
>
> In summary, we believe we have conducted all feasible theoretical analyses and that our hypotheses are validated by empirical results.
>
>
> **[2. Experiments]** Regarding generalization:
> - We would like to emphasize that we primarily focus on targeted data poisoning attacks on image classification models, where gradient-based attacks (such as gradient matching) are performed. Such attacks (e.g., Geiping et al., 2021) are conducted in the setting of CIFAR-10/TinyImageNet with ResNet-18/VGG-16, and we directly adopt their settings.
> - While we appreciate the reviewer's suggestions on multimodal systems, we are not aware of existing gradient-based targeted attacks on such models (as text poisoning using gradients is difficult).
> - Moreover, we believe that even in our setting, poisoning attack difficulty is poorly understood, and we believe such a study is valuable.
> - We would like to draw your attention to our Appendix E, where we consider targeted attacks on generative models. However, as the threat model is vastly different, our proposed tools do not directly transfer.
>
> **[3. Quantitative scales]** We thank the reviewer for their suggestions on thresholding $\delta$ and $\tau$. We wish to clarify that:
> - These two metrics are used to further differentiate between poison classes $y_p$ ​where EPA falls short. As a result, they are not used independently but rather to further rank the poisoning difficulty within groups of targets with similar EPA (line 315).
> - As visualized in Table 3, the thresholds vary between different classes $y_t$​, making it difficult to determine a universal threshold. As a result, we recommend computing $\delta$ and $\tau$ to determine the most dangerous poison classes to trigger an alert for defense or detection (see Appendix A, lines 704-712).
>
> **[4. Anomalies]** To confirm the robustness of our algorithms, we have performed an experiment on calibrated predictive utility to predict the success of targeted data poisoning attacks (GM) on CIFAR-10. We observe an AUC (area under the curve) of 0.93 without specifying the poisoning label $y_p$​ (using only EPA) and accuracies of 87.5% and 88.2% for identifying the poisoned $y_p$​ using $\delta$ and $\tau$, respectively.

---

> > ### Author Response · Authors · 2025-11-28
> >
> > Dear Reviewer t48Y,
> >
> > Thank you again for your detailed and constructive feedback, which has been invaluable in improving our paper.
> > As the discussion period draws to a close, we wanted to follow up to ensure our rebuttal fully addresses your concerns. If any questions or uncertainties remain, we would be happy to provide further clarification.
> >
> > We believe our revisions and responses have meaningfully addressed the issues you raised. If you find our rebuttal satisfactory, we would greatly appreciate if you would consider adjusting your rating accordingly.
> > Thank you for your time and thoughtful engagement with our work.
> >
> > Best regards,
> >
> > Authors

---

### Meta-Review · Area_Chair_XErJ · 2025-12-13

**Summary:**

Reviewers agreed that the paper studies an important problem—instance-level vulnerability in targeted data poisoning—and provides extensive empirical evidence that attack difficulty varies significantly across test samples. The proposed metrics (EPA, poisoning distance, and budget lower bound) are intuitive and empirically correlated with attack success.

However, reviewers raised concerns regarding the degree of conceptual novelty, noting that several components of the proposed approach build upon existing intuitions from prior work on training dynamics and example difficulty. The theoretical grounding of the poisoning distance metric was viewed as largely heuristic, with its connection to actual poisoning optimization processes remaining indirect. Reviewers also questioned the practical applicability of the proposed metrics, particularly the computational cost associated with EPA and the reliance on assumptions such as access to clean labels and training data. In addition, the experimental validation is primarily confined to image classification tasks, which limits the strength of the paper’s generalization claims. Finally, while there exists a substantial body of work on poisoning sample selection that explicitly studies the varying importance of different poisoned samples, the paper does not sufficiently position or discuss its contributions in relation to this line of research.

**Reviewer Concerns:**

Concerns addressed by the rebuttal.
The rebuttal clarified the motivation and scope of the paper, emphasizing that the goal is predicting instance-level poisoning difficulty rather than proposing new attacks. It strengthened the empirical justification of EPA, δ, and τ by providing additional explanations and pointing to consistent correlations across multiple attack methods, training settings, and datasets. The authors also clarified the role of poisoning distance as a proxy metric and acknowledged its heuristic nature, which helped address concerns about misuse or overclaiming. Minor presentation issues and questions about experimental design were largely resolved.

Concerns still outstanding.
Some core concerns remain insufficiently addressed. In particular, questions about the conceptual novelty relative to prior work on training dynamics and example difficulty persist, as the rebuttal primarily reframed the contributions rather than introducing new theoretical insights. The theoretical grounding of the poisoning distance metric remains limited, with no formal link to actual poisoning optimization beyond empirical correlation. Additionally, practical deployment concerns—such as the high computational cost of EPA and assumptions about access to clean labels or training data—remain largely unresolved. Finally, generalization beyond image classification and the qualitative treatment of generative models were not substantially strengthened.

**Reviewer Scores:**

eviewer t48Y.
This reviewer was raised concerns about novelty and practicality. The rebuttal clarified the paper’s scope and addressed several technical questions, which may have moderately alleviated concerns. The score would likely remain the same or increase slightly (e.g., from weak reject to borderline).

Reviewer EM8p
This reviewer focused on empirical validation and clarity. The rebuttal adequately addressed most of their questions regarding experimental design and interpretation of results. As a result, the reviewer might slightly increase their score or maintain a weak accept/borderline position.

 Reviewer cVWR
This reviewer expressed stronger concerns about conceptual novelty, theoretical grounding, and real-world applicability. While the rebuttal acknowledged these limitations, it did not fundamentally resolve them.  As a result, the score would likely remain the same

---

### Decision · Program_Chairs · 2026-01-26

Reject